# Development of Smart Irrigation Equipment for Soilless Crops Based on the Current Most Representative Water-Demand Sensors

**DOI:** 10.3390/s23063177

**Published:** 2023-03-16

**Authors:** Francisco Sánchez Millán, Francisco J. Ortiz, Teresa C. Mestre Ortuño, Antonio Frutos, Vicente Martínez

**Affiliations:** 1Riegos y Tecnología SL. Circunvalación, S/N, 30880 Águilas, Spain; 2Department of Automation, Electrical Engineering and Electronics Technology, Universidad Politécnica de Cartagena, St. Dr. Fleming, s/n, 30203 Cartagena, Spain; 3Centro de Edafología y Biología Aplicada del Segura, CSIC, Apdo 164, 30100 Murcia, Spain

**Keywords:** soilless cultivation, smart irrigation, precision agriculture, gravimetric, volumetric water content (VWC), data acquisition, gravimetry (G), accumulated solar radiation (AR), soilless culture system (SCS), growing media (GM), crop evapotranspiration (ETc), water use efficiency (WUE)

## Abstract

Due to the edaphoclimatic conditions in southeast Spain, which are expected to worsen due to climate change, more efficient ways of using water must be found to maintain sustainable agriculture. Due to the current high price of irrigation control systems in southern Europe, 60–80% of soilless crops are still irrigated, based on the experience of the grower or advisor. The hypothesis of this work is that the development of a low-cost, high-performance control system will allow small farmers to improve the efficiency of water use by obtaining better control of soilless crops. The objective of the present study was to design and develop a cost-effective control system for the optimization of soilless crop irrigation after evaluating the three most commonly used irrigation control systems to determine the most efficient. Based on the agronomic results comparing these methods, a prototype of a commercial smart gravimetric tray was developed. The device records the irrigation and drainage volumes and drainage pH and EC. It also offers the possibility of determining the temperature, EC, and humidity of the substrate. This new design is scalable thanks to the use of an implemented data acquisition system called SDB and the development of software in the Codesys programming environment based on function blocks and variable structures. The reduced wiring achieved by the Modbus-RTU communication protocols means the system is cost-effective even with multiple control zones. It is also compatible with any type of fertigation controller through external activation. Its design and features solve the problems in similar systems available on the market at an affordable cost. The idea is to allow farmers to increase their productivity without having to make a large outlay. The impact of this work will make it possible for small-scale farmers to have access to affordable, state-of-the-art technology for soilless irrigation management leading to a considerable improvement in productivity.

## 1. Introduction

The UN’s Food and Agriculture Organization (FAO, Rome, Italy) estimates the world population to be 9.6 billion by 2050 [1]. The sustainability of irrigated agriculture in the semi-arid Mediterranean region is threatened by the overexploitation of natural resources and changes in the use of agricultural land. Agriculture is the dominant water consumer in the region, accounting for 81% of the total water consumed, and is particularly vulnerable to climate conditions due to its dependence for most of the year on a plentiful supply of good-quality water [2]. Climate change is expected to have a major impact on water availability and supply, infrastructures, and farmers’ incomes; it also leads to reduced production and increased food insecurity, with social and economic consequences involved, such as higher levels of poverty and migration [3].

Population growth, urban development, and industrial progress will be difficult to sustain without proper support for agriculture, which consumes most of the world’s available water resources [4]. For example, in Europe [4], the United States [5], India [6], Ethiopia, and South Africa [7], agriculture accounts for about 60% of the total social water consumption, more than 90% in some regions. Water-saving technologies (WST) in agriculture, developed through technological development applied to the management of available water resources, are encouraged in many areas [8]. Although improved irrigation and water-saving practices have been adopted by some growers, they have not been widely adopted since the growers do not benefit directly from saving water, and these systems are expensive. Studies show that low adoption rates are an important factor in restricting the effectiveness of WST [9]. Increasing farmers’ WST adoption rates undoubtedly has a positive impact on increasing yields and income [10]. Saving water resources is also conducive to improving the ecological environment and maintaining sustainable economic and social development [4]. Mediterranean countries have been identified globally as climate change “hot spots” in which heat waves are expected to increase by 200 to 500% due to greenhouse gas emissions [11]. Indeed, irrigation demands are relatively high, especially in arid and semi-arid areas, due to the high radiation load throughout the year, but this is now also expected to occur in more temperate and sub-tropical zones [12]. Tomatoes (*Solanum lycopersicum* L.) are one of the most important crops worldwide [13] and are of great interest to today’s society as they are a source of compounds with great health benefits [14]. According to the FAO [15], Spain is the second-largest tomato producer in Europe, with approximately 56,000 hectares under cultivation and a yield of 85 t/ha.

Farming land, soil, and water are the fundamental resources required for conventional agriculture. The first step towards the optimal use of these resources is the incorporation of soilless farming techniques [16]. With these methods, plants can grow in an indoor environment, reserving the land for some other use. Soilless culture systems (SCS) in an almost completely controlled environment constitute a relatively modern cultivation technology and are used almost exclusively in greenhouses. Cultivation on horticultural growing media (GM) such as rock wool, perlite, and coconut are the most frequently used SCS for the production of fruit, vegetables, and cut flowers all over the world [17]. Attempts have long been made to eliminate the problems associated with soil in greenhouses, including soil-borne diseases, low soil fertility, and high salt levels. Over the past 30–40 years, the development of suitable growing media with optimal physical and chemical properties has led to soilless cultivation in greenhouses taking a leading role. This development has also been favored by advances in plant nutrition and irrigation through modern fertilization approaches and automation [18].

In arid and semiarid regions, the demand for water is constantly increasing as freshwater resources are being exhausted. However, more water is needed to meet the future demands of food production. The best solution is thus to develop a water-saving management technique and/or move to adopt non-conventional sources of water for crop production. The ideal solution would be to promote the use of marginal quality water, especially saline water, for agricultural purposes [19].

However, if saline water is to be used on a large scale for crop irrigation, it must be based on proper irrigation management, and this is much more complicated than that of freshwater [20]. Many of the problems in irrigated soil can be controlled in soilless crops, so there is considerable potential for employing saline water irrigation in soilless media, particularly for the production of vegetables [21]. Using saline water for irrigation involves a higher leaching fraction in soilless crops, which affects plant nutrition and physiology [22]. Irrigation systems should thus consider the quality of water for irrigation management.

Different irrigation methods have been described [23], such as occasional irrigation, pulse irrigation, high-frequency irrigation, and continuous irrigation. These authors also describe several approaches to making irrigation decisions: look-and-feel, gravimetric, timer-based, sensor-based, and model-based methods. In the current state of research on automatic irrigation systems in soilless crops, various systems can be found, many of them commercial products already in operation. Some irrigation scheduling methods are based on the application of a water balance and require an accurate estimation of the plants’ water consumption. Crop evapotranspiration (ETc) can be estimated by indirect (climate-based) methods or direct methods (using weighing or drainage lysimeters, balances, etc.) [24]. The indirect methods of ETc calculation require climate sensors for radiation, temperature, and humidity. Direct methods include gravimetric sensors [25], measuring the moisture of the substrate, using granular matrix sensors or capacitive sensors [26], or measuring the water potential or matrix voltage by tensiometers or electro-tensiometers [27]. An optimally designed irrigation system will deliver water to the plants to maximize water use efficiency (WUE) [23]. A detailed review of the most representative irrigation methods for soilless crops has been performed [28].

For scheduling the irrigation of soilless crops, 60–80% of the crops in Italy, France, Poland, Greece, and Hungary are irrigated based on the experience of the grower and/or advisor. Methods that either calculate ETc or are based on incoming solar radiation are used in 60–95% of the crops in The Netherlands, the United Kingdom, Portugal, and Belgium. Demand trays are the predominant method in Spain [29]. Weighing balances are used in 25% of Dutch soilless crops. Soil/substrate moisture sensors (including tensiometers) are used in 10–15% of soilless crops in Spain, Italy, The Netherlands, and Portugal. These data show that there is considerable room for improvement in the irrigation management of soilless vegetable crops [30].

In all irrigation systems, decisions must be made about when to irrigate and for how long. In soilless growing systems, the irrigation process can last a few seconds or as long as several minutes. Irrigation schemes are management strategies designed to attain specific crop production goals through various delivery and monitoring methods. Optimal irrigation management is required to appreciably enhance water productivity (or water use efficiency) for crops produced in greenhouses [31].

Agriculture 4.0 has been triggered by emerging technologies such as the Internet of Things (IoT), artificial intelligence, data analytics, blockchain, big data, etc. [32]. These technologies address food safety-related issues, modernize the agricultural supply chain [33], improve data representation, storage, and analysis, and provide better management and control.

Finding a new way to monitor single-plant water consumption is important to better determine how much water should be provided to improve crop yield, quality, and so on. Some authors have explored the relationship between the daily irrigation applied and melons’ daily weighted ET (WET) and then calculated the appropriate amounts by weighing the plants [34]. A new, low-cost weighing lysimeter has been developed for potted plants and its technical and economic evaluation in a 16-unit lysimetric water consumption analysis station [35].

After studying the current state of the art and the existing commercial systems, we detected a market requirement for a low-cost, scalable control system able to work together with any type of fertigation equipment. The systems currently available on the market are considered expensive and require a large investment, so the technology is only available to large-scale producers, leaving out the small farmer. As they can only work with an irrigation controller of the same brand, the cost of expansion to multiple zones skyrockets due to the software and wiring required. The hypothesis of this work is that the development of a low-cost control system with high performance will allow small farmers to improve the efficiency of water use by obtaining better control of soilless crops. Therefore the aim of this work was to develop a low-cost system for self-managed irrigation of soilless crops after selecting the most efficient method of the three most representative irrigation systems and considering the quality of the irrigation water. 

The new system is scalable and able to work with any of the fertigation controllers currently on the market. The impact of this work will make it possible for small farmers to have access to affordable, cutting-edge technology for soilless irrigation management with considerably improved productivity.

## 2. Materials and Methods

Three irrigation control methods were compared in a study to collect data for the development of an intelligent irrigation monitoring and control system. Our aim was to determine the best sustainable system of obtaining higher yields together with efficient use of water. 

### 2.1. Agronomic Trials

The experiment was carried out in a greenhouse in the experimental farm “Tres Caminos” of the Centro de Edafología y Biología Aplicada del Segura (CEBAS) in the municipality of Santomera (Murcia), whose coordinates are 38°06′26.32″ N and 1°02′07.14.14″ W (Figure A1).

Agronomic trials were carried out with the Sigfrid variety of tomatoes (Solanum lycopersicum) to evaluate three different irrigation control systems for soilless crop production: Accumulated solar radiation (AR); Gravimetry (G); and Volumetric Water Content (VWC) with two types of water: Control (C, “Hoagland” Nutrient Solution); and Salinity (S, “Hoagland” Nutrient Solution with 50 mM NaCl). The trials had a bifactorial design (3 × 2) with three irrigation methods combined factorially with two water qualities in a random block distribution. In each treatment, 4 culture bags were placed per block, with 3 plants each, which made a total of 6 treatments and 24 plants (replicates) per treatment. Planting density was 2.5 plants/m^2^. The drip irrigation system consisted of a 2 L/h non-draining and self-compensating dripper per plant. Treatments C-RA, C-G, and C-VWC were irrigated from a single control tank, and S-RA, S-G, and S-VWC from another tank containing a saline solution. The nutrient solution pH and EC were monitored, and drainage EC was measured periodically. This distribution can be seen in Figure 1.

The 100 × 18 × 16 cm culture bags were made of cocopeat, and the cocopeat substrate had the following composition: organic material 92%, ash 8% m/mS, contraction <15% *v*/*v*, dry bulk density 69 g/L, pH 5.5–6.5 and EC < 2.5 mS/cm.

#### 2.1.1. Analytical Data

The plants were allowed to grow until production. Tomato fruits were harvested daily when they turned red. The fresh weight and size of the individual tomatoes were determined by the plant and treatment. The fruits were classified into commercial and non-commercial categories (small for weight less than 50 g or having some physiopathology such as cracked or blossom-end-rot).

#### 2.1.2. Statistical Analysis 

The statistical analyses of production were carried out by two-way ANOVA (SPSSv27 Statistical Package, Chicago, IL, USA) with water quality and irrigation methods as the main factors, using a *p* = 0.05 probability cut-off as the indicator of significant differences. The values significantly different at 95% were separated according to Duncan’s multiple range test.

### 2.2. Irrigation Control Systems: General Architecture of the Test System

A SCADA System was created in a PC that communicated via Ethernet-IP with a PLC in the General Control Panel to monitor the installation and obtain irrigation data. The Main Control Box was connected by cable with the individual Tray Control Box of each area, in which different transmitters were installed, as shown in Figure 2.

Both irrigation and drainage volumes were measured to adapt the irrigation dose to obtain optimal drainage according to the water needs of the plants. EC and pH were also measured in each of the irrigation sectors. Measuring parameters such as EC and pH in the drainage and comparing them with those of the irrigation water indicates the nutritional evolution of the crop and helps to monitor the nutrient solution.

The main control panel consisted of a Siemens Simatic S7-1200 Series, Model 1214C DC /DC/DC CPU. This PLC was programmed to control the three irrigation methods individually. The control panel was connected by cable with the individual control panels in the weighing trays, with radiation and VWC sensors, and with a distribution panel that sent the activation signals to the solenoid valves in each sector. 

#### 2.2.1. Volumetric Water Content (VWC) Control System

This simple system consisted of an on-off control to activate irrigation when a certain volumetric water content is detected in the substrate below a defined set point and stop it when the limit is reached. A MAS-1 model sensor (Decagon Devices, METER Group, Inc., Pullman, WA, USA) was used to measure the volumetric content of the substrate, operated by measuring the dielectric constant of the substrate to determine its volumetric water content, which is very sensitive to water content. The sensor supplies a 70 MHz oscillating wave, which induces an electromagnetic field in the substrate around the sensor. A microprocessor in the sensor measures the dielectric constant of the substrate, which is related to its volumetric content. The microprocessor measures the dielectric and updates the current once per second. The transmitted 4–20 mA current is converted to volumetric substrate content by the following correlation in the control software using rock wool as substrate:VWC = 0.00446 × mA^2^ − 0.0359 × mA + 0.0741(1)

The scale 0–100 (% VWC) is expressed as a percentage of Span (FS). Applying these equations will generally result in an accuracy of ±4% VWC as long as the electrical conductivity of the medium is less than 8 dS/m. The system is based on activation below a VWC Set Point and a stop above a set maximum VWC value.

#### 2.2.2. Radiation (AR) Control System

Solar radiation is the most important energy source for a plant and a determining climatic factor for crop development and yield. Low radiation levels lower the chlorophyll content and reduce the plant’s photosynthetic activity and photochemical concentration. Solar radiation determines the demand for water vapor in the atmosphere and is, therefore, a condition for the crop’s irrigation needs. Irrigation is initiated when the energy set in Wh/m² is reached, restarting the accumulated radiation counter (0 Wh/m²) and continuing until the set point is reached again.

The Apogee SP-110 sensor pyranometer was used in the AR-controlled areas. This had a supply voltage of 12 Vdc, a sensitivity of 0.20 mV per W·m^−2^, and an uncertainty calibration of 5%. It was mounted on a level horizontal surface and faced north to minimize Azimuth error. The sensor was connected to a transmitter capable of measuring the signal in a range of 0–250 mV. The input range was adjusted to the pyranometer output range to maximize signal resolution and minimize noise. The pyranometer can be used to find the instantaneous value of the solar irradiance in W/m^2^ or by integration to find the total energy per square meter, which strikes the instrument during a given period. The calibration factor K can be expressed as follows:K = 200 mV/(kW/m^2^)(2)

This value means that when the solar irradiance equals 1 kW/m^2^ (typical for a clear, sunny day around noon), the pyranometer output voltage will be about 200 mV. If the output voltage, for example, is 100 mV, then the solar irradiance would be about 0.5 kW/m^2^ = 500 W/m^2^. In other words:S[kW/m^2^] = U[mV]/K(3)
where U is the signal voltage in millivolts, and S is measured in kW/m^2^.

#### 2.2.3. Gravimetric Control System

For the treatments controlled by the gravimetric method (G), 4 weighing trays were placed on the crop line, following the continuity of the channel. The drainage from the weight trays was first measured by the double bucket counter equipped with a reed contact, drained through the filtered drains at the base of the collector, and then fell into the channel to be counted by another counter at the end of the line together with the drainage from the other three bags.

After the drainage has been completed, the weight variation of the tray defines the plant’s water needs and, unlike the (VWC) and (AR) systems, is a direct measure of the state of the crop. When the weight variation is equivalent to the volume consignment consumed by the Tray System, a new irrigation cycle will begin.

For the weight measurement, two traction load cells in each tray converted the load acting on them into electrical signals proportional to the weight. For maximum sensitivity and temperature compensation, two strain gauges under tension and two under compression were connected to a Wheatstone bridge. 

Knowing the power supply voltage, the force value was quantified by connecting both load cell outputs to a box with the 4–20 mA signal conditioning and transmission circuit. Since the output voltage in each load cell is proportional to the variation of the resistances in the Wheatstone bridge, and this, in turn, is proportional to the linear variation of the weight on the platform, a 4–20 mA output signal with linear variation with respect to the weight on the tray is obtained.

All the load cells receive the same power supply, and when they are “unbalanced” by the effect of the weight of the product, each one supplies the same mV signal. Since the cells are in parallel, the end result is the mV signal. The tray was aligned so that the same weight was supported by each of the “S” type load cells, in this case, 40 kg, so the weighing system can support 80 kg, including the dead weight (tray where the load is placed) and the maximum weight of the load. An amplifier was used to condition the output of the load cells.

It is very important to calculate the error measurement for system validation. To do this, the linearity error of the load cell was taken into account, which is expressed as a percentage of span (FS), or the difference between the upper and lower measurement limits, and which, in this case, has a value of 0.03% FS. To calculate the error of the analog conditioning system, the “Accuracy” value offered by the transmitter’s table of technical parameters was used and was 0.3% FS (Span). By adding both errors, it was possible to calculate the total error, both in the most unfavorable case Ɛ*_TMD_*, and in the most likely Ɛ*_TMP_*:Ɛ*_TMD_* = Ɛ*_lin_* + Ɛ*_RW_*_−*ST*01*A*_ = 0.03 + 0.3 = 0.33%*FS*(4)
(5)$${Ɛ}_{TMP}=\sqrt{{Ɛ}_{lin}{}^{2}+{Ɛ}_{RW-ST01A}{}^{2}}=0.30\%FS$$

The transmitter was found to be correctly set to calculate the error by using known weights. For this, zero (4 mA) was obtained by disregarding the dead weight of the tray, and the Span (20 mA) 80 kg was considered. Checks were then carried out by taking ten measures with known weights in ascending order and ten in descending order.

The actual results obtained were consistent with the theoretical ones, and the errors obtained were admissible. A sensitivity test was also carried out in which it was found that the weight measurement system, consisting of a load cell and transmitter, was capable of detecting weight increases of 5 gr. As we approached the upper range limit, the minimum detectable weight increases was 10 gr. These results were also admissible. To eliminate noise at the output, twisted and shielded pairs were used to transmit the signal (Table 1).

#### 2.2.4. Control Software Development and SCADA

Software integrating the control algorithm of the three decision methods was developed instead of individual independent controllers. The KOP language was used for the software development in a Siemens TIA PORTAL environment. A GRAFCET diagram showing the control flow of the program can be seen in Figure 3.

Irrigation volume and drainage were established for the three control systems, as well as the number of plants, so that the system was able to calculate the total irrigation volume:V Total irrigation (L) = Num plants × Vol per plant × (1 + 1/% Drainage SP)(6)

For the method (VWC), after calibrating the volumetric water content sensors, it was established that the difference between the saturated substrate after irrigation, and with 1.5 liters less, corresponded to a volumetric water content difference of between 50% and 60%. Irrigation began when volumetric water content fell below 50% until reaching the calculated volume.

In method (G), irrigation was initiated when the system was initially activated and delivered the established volume when the system obtained a reference weight so that when the weight loss of the tray is equivalent to the consumption of its plants, a new irrigation cycle is initiated. In this way, a consumption volume was established in which drainage was not considered:V Tray (L) = Num plants × V per plant(7)

For radiation control (R), a relationship was established between the accumulated radiation and crop evapotranspiration, i.e., the water consumed by the crop. The water consumed was calculated by the difference between the irrigation volume and subsequent drainage. With these data, the accumulated radiation was established, which, when reached, would initiate irrigation and put the accumulated radiation back to 0 again. A SCADA System was developed in the Siemens TIA Portal environment, with communication via Ethernet-IP with the PLC to obtain the configuration of the installation and data from the tests carried out, as shown in Figure 4.

## 3. Results and Discussion

### 3.1. Comparative Agronomic Study of Irrigation Control

Agronomic trials were carried out in a greenhouse to determine the most appropriate irrigation control model, using two types of water and three systems for monitoring the crop water status (gravimetric, accumulated radiation, and volumetric water content sensor). Different agronomic parameters were determined to validate both the calculation systems and the accuracy of the measurement systems. Under the control conditions, the irrigation method significantly affected the yield, the gravimetric method being the one that obtained the greatest commercial yield. The other two methods used obtained the same production, below the gravimetric method. This difference in production could be explained by the gravimetric method being adjusted more precisely to the plants’ water needs.

The impact of different irrigation methods could be related to the water availability in the tomato plants’ different phenological stages [36]. According to [37], increasing soil moisture during the last three stages (fruit growth, FG; fruit development, FD; and fruit maturity, FM) could significantly increase tomato yield. Excessive and insufficient soil moisture during the fruit setting (FS) stage had negative effects on tomato yield. The commercial tomato yield is significantly reduced by water deficiency during flowering and/or yield formation phases [38]. Under saline conditions, the irrigation method also significantly affects the commercial yield. As can be seen in Figure 5, salinity drastically reduced the commercial tomato plant yield by 68%.

A statistical analysis did not indicate any significant interaction between irrigation methods and water quality treatment, so the effect of the different methods on yield did not depend on the quality of the irrigation water. The gravimetric method obtained the highest yield, as shown in Figure 5.

Both the quality of water and the irrigation control system used had an influence on the study of the number of fruits and fruit size (commercial category) (Figure 6). The interaction between the two factors was significant only for fruit size (*p* < 0.05; *). It is worth noting that, once again, the gravimetric control system obtained the highest number of fruits (Figure 6A), while salinity reduced the number of fruits drastically compared to the control conditions (Figure 6B). No difference was obtained between the irrigation methods in terms of the average size of the fruits under control conditions. With salinity, the gravimetric system obtained a larger fruit size (Figure 6C). The reduction in commercial yield produced by the control conditions was mainly due to a decrease in the number of fruits, with no significant effect on the size of those obtained. The reduction in commercial yield produced by salinity was mainly due to the reduced fruit size and the number of fruits obtained (Figure 6).

Our findings indicated that salinity had a negative impact on the production of commercial fruit, regardless of the irrigation method used. Salinity had an impact on commercial production, lowering both fruit size and fruit yield (Figure 5 and Figure 6). Under longer and more intense periods of stress, fruit weight often decreases [39]. According to other studies, under high salinity, a decline in yield can be due to a decrease in fruit weight and fruit production.

Conversely, a decrease in fruit size is the primary factor reducing fruit yield in conditions of low salinity [40]. The cultivated tomato is considered to be “moderately sensitive” to salinity, which indicates that it can withstand an EC of up to 2.5 dS/m without a significant yield drop, but any slight increase in irrigation water salinity is expected to result in production losses [41].

The irrigation method significantly affected the non-commercial yield percentage in both control and salinity (Figure 7). The gravimetric method had a lower percentage of non-commercial fruits. A statistical analysis indicated a significant interaction (*p* < 0.01; **), so the effect of the different irrigation methods on the percentage of non-commercial yield depends on the quality of the water used for irrigation. Figure 8 shows how the influence of salinity raised the non-commercial yield percentage by more than 20% across all the irrigation methods. The lowest amount was obtained by the gravimetric approach. In other studies, at least 10% of the total fruit production could not be commercialized because the fruits had various physiological issues, such as BER, cracking, or were too small (non-commercial yield). This fraction increased to 20% when the salinity of the nutrient solution increased [42]. The transport of calcium from the soil to the fruit is impeded by a shortage of soil water during the fruit’s development and ripening stage, which leads to blossom-end rot and lower yields [43].

Depending on the irrigation technique used, the proportion of non-commercial tomatoes varied and was smaller when the tomatoes were irrigated by the gravimetric method (Figure 7). Other authors claim that changes in the amount of water available to the plant by the irrigation method adopted have a significant impact on the commercial tomato output. According to [40], increasing water shortages in the root zone indicate that marketable fruit yield losses are proportionally greater than the reduction of the water used by the crops. According to [44], when applied continually, instead of only during such phenological phases, the water deficit reduces water use but can have a negative impact on fruit yield. Tomatoes require abundant water from the FS stage to the FM stage, especially during the FD and maturity stages, when water is a growth-restricting factor [35].

From the data obtained during the tests, it can be concluded that the best control system of the three under study was the gravimetric method in terms of yield. According to the good results, the gravimetric method seems to be the best adapted to the irrigation management of the cocopeat substrate since it seems that it was the method that improved the availability of water of the substrate in relation to the plants’ water needs. When the irrigation method was VWC, the production results were worse than those of the gravimetric method, which could have been due to possibly imprecise measurements on some occasions, as well as some failures detected in the sensor measurements of the sensors throughout the crop cycle. With the accumulated radiation method, the commercial yield results were similar to those of the VWC method, which was not successful because the plants continued to transpire at night when the radiation could not be accumulated. This method would, therefore, not be compatible with the water needs of the plants.

### 3.2. Final Development of a Commercial Prototype of Gravimetric Control System

From the data obtained during the tests, it can be concluded that gravimetric control is the most efficient method of improving crop yield. For this, a more affordable commercial prototype smart tray was developed with numerous improvements over the current commercial systems as regards both hardware and software, thanks to the experience gained during the agronomic trials.

#### 3.2.1. General Architecture of the Gravimetric Control System

To reduce costs and create highly competitive equipment, we decided to work with a Schneider PLC of the 241 series, model TM241CE24T/U, in the Codesys programming environment. The main reason for changing the PLC model was the possibility of digital communication via the Modbus RTU protocol without incurring any additional costs.

In this development, the PLC communicates through a switch with an HDMI screen and through the Modbus RTU protocol with a data acquisition card that will collect information from all the sensors in the control zones. This reduces wiring costs and simplifies assembly, as it is no longer necessary to send individual signals to the control panel, as when using 4–20 mA signals.

A 7″ Weintek MT807oiE HMI screen was chosen instead of the PC-based SCADA system due to its features and low price and for the following reasons:-It provides remote access to the program and to data, graphs, and historical data and can download data, which requires a license at a very low cost.-It can even act as a gateway to access the PLC programming itself and even modify it if necessary.

As depicted in (Figure 8), in this case, the control equipment can give an activation order to any of the currently available fertigation devices through an operations panel, which can be any type of device (mobile, PC, tablet) with an internet connection.

#### 3.2.2. Improvements in Electronic Instrumentation and Communication

Tensile load cells were used for the tests. This presented a structural problem as the cell supports interfered with the plant support, so torsion load cells were used in the commercial model.

Another common problem that arose during the tests in equipment with low flow meters was related to the magnetized double spoon counters with reed contacts to measure the drainage volume. Other systems of pot cultivation use a conduit crossing the platform connected to a spherical PVC drainage tank. This container was suspended from another loadcell which weighed it and the water it contained. Once the tank was full, two electro-valves (one connected to the input and the other to the output) took over to open or close the tank in such a way that no drainage water was lost in the process [35]. Ref. [45] used load cells for direct estimation of irrigation and drainage water in soilless systems. As cited in [46], measurement systems range from weighing the lysimeter measuring output every 10 min to calculating a 60 min average of 1 min measurements, although the expense of these systems limits their use to research on plants grown in containers.

A reed relay has a mechanical switching capacity of millions of operations; its service life is conditioned by its load, which can be a maximum of a few tens of mA with a maximum voltage of tens of volts. The fact that it was installed switching a 24 Vac relay with no RC in either the relay coil or the reed relay itself may have been the cause of the problems. Although the magnetization in the double spoon counters was mainly due to voltage peaks caused by the intermediate relays, mitigated by solid state relays, it was decided to use another technology to obtain a permanent solution, as the company’s export profile requires robust solutions that require little maintenance.

Figure 9a shows the voltage peaks caused in a reed contact when an intermediate relay is used. When a solid-state relay was used (Figure 9b) the voltage peaks were considerably reduced.

Despite the reduction of voltage peaks, as the reed contact has a limited number of operations, a new technology based on the Hall effect was developed. The Hall effect is based on the appearance of an electric field due to the separation of charges inside a conductor through which a current flows in the presence of a magnetic field with a component perpendicular to the movement of the charges. This electric field is perpendicular to the movement of the charges and to the perpendicular component of the applied magnetic field. A Hall-effect sensor, being a semiconductor with no mechanical elements, is thus capable of an unlimited number of maneuvers. The newly designed sensor, named the SCE-CCHR1, is a circuit with a Hall-effect sensor to detect the passage of a magnet installed in a spoon. In SCE-CCHR1, the sensor attacks the gate of a self-protected MOSFET, so the load can perfectly well be hundreds of mA over tens of volts and remain in continuous operation. This new prototype also includes the following features:-Polarity reversal protection.-Linear voltage regulator.-Self-protected Mosfet output.-Transistors at the input and output for transient protection.

Hall sensors of different sensitivities were tested during the development, and we chose either the one with the highest sensitivity or the one whose signal (period and pulse width) was closest to that delivered by the original spoon from the available models. Figure 10b shows the signal obtained by the most sensitive Hall effect sensor.

A “CCHR1 tester” was then developed, which allowed thousands of pulses to be performed without using water, and the prototypes were short-circuited. We then used an electromagnet obtained from a Releco relay to excite the double spoon sensor thousands of times without using water. The electromagnet is activated by an ESP32 installed on an SDB1 board, which can also read the pulses at the CCHR1 output. The number of pulses, period, and width of the pulse is selected by a preselector. Finally, new software was developed for this tester.

Tests and measurements were carried out to check the functionality of the new counter with both the original double spoon counters with a reed switch and the SCE-CCHR1 with a Hall effect sensor in two different situations, real conditions with water and with the CCHR1 tester. The output of both devices was connected to different types of charges:Digital input of SDB1, equivalent to PLC’s NPN digital inputs.24 Vdc relay without protective circuit against overvoltage.24 Vdc relay with different RC’s protective circuits.24 Vdc relay without 1N4007 freewheel diode.

All these tests were monitored by a digital oscilloscope with the probe applied to the load so that the signals were recorded. The following events were observed:Voltage peaks of hundreds of volts at the 24 Vdc relay without a protective circuit.Peaks were reduced when RC protective circuits were used.Negligible overvoltage with the 24 Vdc + 1N4007.The SCE-CCHR1 with a 24 Vdc + 1N4007 relay load or digital input of the SDB1 gave signals with no overvoltage.

Pulses with a period of 350 ms and a width of 80 ms were measured in real conditions, both with the reed sensor and with the Hall effect sensor, equivalent to about 3 pulses per second. Then, 5ms pulses were applied with the tester every 10 ms, corresponding to 100 pulses per second. Under these conditions, the reed relay was not able to respond at this speed, but the Hall effect relay was, with millions of pulses being applied to the Hall effect relay during the tests.

Another disadvantage found during the tests was that the large number of sensors meant that complex wiring was necessary, which entailed a high cost in terms of electrical wiring and assembly. Other developments have opted for dataloggers with multiplexers to increase the number of input channels [35]. Artificial neural networks have been applied in agricultural systems mainly for open field cultivation to estimate soil moisture content based on various soil and environmental parameters and also for irrigation planning [47].

This complex wiring and cabling have a high economic impact when the tray is at some distance from the service area and/or when there is more than one control zone. To reduce costs and make the system scalable, a data acquisition board, called the SDB (Tray System), was designed (Figure A2). This SDB includes the load cell transmitter, the pH, and the EC drainage transmitter, which is able to read several substrate sensors simultaneously with the SDI-12 protocol and also receives signals from the irrigation and drainage meters. Communication between the SDB and the controller is via Modbus RS-485 RTU with only four wires.

If the installation has more than one control zone, it will be possible to work with a parallel connection by means of the Modbus-RTU protocol. This will guarantee the scalability of the equipment and make it possible to increase the number of control zones in an installation at a reduced cost.

Since the control panel is in the service area and the SDB in the greenhouse, to simplify sensor calibration, the option of using a mobile application via Bluetooth connection is offered, which can also be used to update the software if necessary.

When communicating with more than one SDB, termination resistors must be used that reduce the sensitivity to electrical noise, with a termination resistance equal to the characteristic impedance of the cable (normally 120 Ω for twisted pairs).

#### 3.2.3. Control System

In order to create scalable equipment, in addition to using SDB modules with Modbus-RTU connection, programming was carried out in the Codesys environment using function blocks and variable structures (Figure A3).

The equipment can thus be adapted to the needs of the individual installation, as the modular structure of the programming makes it easy to expand the software from one control zone to as many as necessary. As in the test control system, the intuitive and visual Ladder language was chosen for programming. For the screen, we opted for an HMI control panel as it has practically all the possibilities of a SCADA system, such as:Visualizing the information and monitoring the processes running in the machine in real-time. As shown in (Figure A4), it is possible to record the values of the irrigation and drainage volumes, the weight of the tray, and its drainage percentage, as well as the irrigation time and the values provided by each of the sensors.Entering the information needed to operate, select recipes, display indicators, etc.Display and control instruments and mechanisms. In this case, the number of plants and the number of drippers per plant, the drainage command, and the counter volumes per pulse must be entered (Figure 11). It is also necessary to define an active schedule as well as the volume consumed per plant.Alarms, permissions, irrigation graphs, and logs (Figure 12). Macros were created to obtain all the information regarding the measured weight magnitudes, EC, pH and drainage temperature, EC, VWC, and substrate temperature and the values of the irrigation and drainage volumes, and other indirectly calculated values such as drainage percentages or plant consumption.

This control system has been thoroughly tested, in the laboratory, in a Mexican commercial fruit facility, and in the South of Spain (Figure 13).

The next steps that can be taken to continue this development are to work on a more complex control system that takes into account other parameters such as drainage EC or the Integral Daily Light (ILD) without increasing the cost of the system, and that is capable of self-configuration, learning from both experience and the data accumulated in the system. Another possible development is the implementation of a via radio communication with the protocol Modbus rtu.

### 3.3. Conclusions

From the data obtained during the agronomic tests, it can be concluded that the best control system of the three under study was the gravimetric method in terms of yield. Judging by the good results, the gravimetric method seems to be the best adapted to cocopeat substrate irrigation management since it was the method that was found to improve the availability of water in the substrate in relation to the pants’ watering needs. After selecting the method with the best agronomic performance, we developed a competitive, cost-effective automatic gravimetric control system based on the experience gained during the field trials of the first prototype. This new system had a significant cost reduction of around 50% less than the current leading commercial systems. It is also scalable, can be remotely controlled, receives software updates, and provides access to all the irrigation data. This new design is compatible with any current fertigation controller through the use of external activations and shutdowns, unlike the costly equipment on the market, which requires a controller of the same brand. The two factors of price and compatibility put this control equipment within reach of small producers, who can increase their production significantly for a small investment.

Valuable know-how was gained from experience obtained during the trials to design gravimetric commercial irrigation control equipment. Although traction load cells were used in the tests, it was found that their supports interfered with the plant trellises, so it was decided to opt for torsion load cells, which do not interfere with the plant. The tray is also aligned with the cultivation line for minimal visual impact. Another problem that arose during the tests was that the contact counters used became magnetized and stopped counting the pulses, which interfered with the control system. A new non-magnetizing Hall-effect sensor was thus developed for flow meters that can withstand an unlimited number of operations, replacing the reed contacts in the double-bucket meters for a slight cost increase. These meters did not become magnetized during the tests, while the error was in no case greater than 5%.

The SDB data acquisition card was developed to receive all the signals from the multiple sensors to reduce wiring and minimize assembly times. This data acquisition system developed for the weighing control system, in addition to receiving information from the pH and EC drainage sensors, the load cells, and the irrigation and drainage meters is able to communicate by the SDI-12 communication protocol with several substrate sensors. These sensors provide information on temperature, volumetric water content, and the crop EC, which is obtained by a correlation according to the crop in question. The information obtained by the SDB is sent via Modbus-RTU protocol to the PLC. A PLC working in the Codesys environment was chosen to enable the use of standard Modbus-RTU communication protocols. Thanks to the SDB, this equipment is scalable because parallel communication can be made with a PLC via Modbus-RTU, allowing a large number of trays to be networked, considerably reducing the wiring requirements and labor costs and facilitating future expansion. The SDB communication sensors can be calibrated via an APP on a mobile phone. This is a significant improvement over other equipment currently available on the market, as the sensors can be calibrated via Bluetooth in direct communication with the SDB data acquisition system. A single operator can thus remotely calibrate the sensors in a tray hundreds of meters away from the service area with the main control panel.

An HMI display was chosen, which offers the same features as an expensive SCADA system, with the possibility of remote control, and which does not charge according to the number of variables used. This HMI interface will also serve to access the PLC programming. Communication between data acquisition cards and PLC via radio is foreseen in future developments, which will eliminate the need for wiring. The fact that the software works in the Codesys environment, based on function blocks and variable structures, means the equipment’s control zones can be increased almost automatically. It should be noted that remote access is available to the programming parameters, to all the irrigation data, and to remote graphics. These features make this new equipment an affordable and efficient irrigation control system for soilless crops, with improved crop yields and savings in water and fertilizers, making it possible for small-scale farmers to obtain a huge improvement in productivity. In the next steps, we expect to increase the complexity of the control so that other parameters, such as drainage EC or the daily light integral (DLI), are taken into account without incurring system cost increases.

Finally, it should be noted that this control system has been thoroughly tested, both in the laboratory and in the test facility itself, and has recently been put on the market with encouraging results while further trials are being conducted to determine how well a weighing tray represents the conditions in a given area.

## Figures and Tables

**Figure 1 sensors-23-03177-f001:**
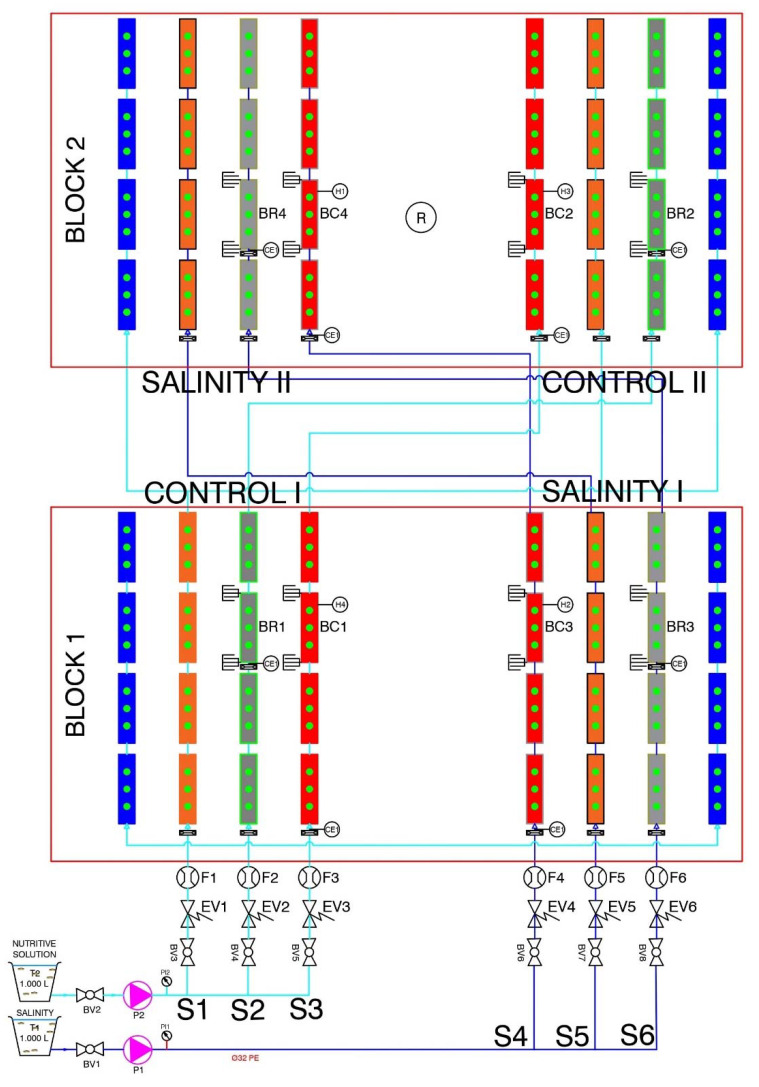
Scheme of the experimental design. Two saline treatments were factorially combined with 3 control systems. Accumulated solar radiation (AR) treatments S1 and S5 are in orange, Gravimetry (G) S2 and S6 in grey, and Volumetric Water Content (VWC) S3 and S4 in red.

**Figure 2 sensors-23-03177-f002:**
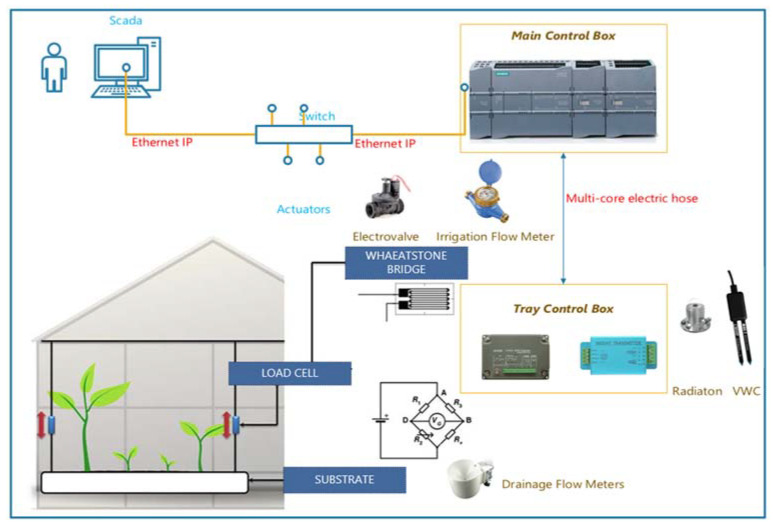
General view of the test facility control architecture to manage three different irrigation methods, Gravimetry (G); Volumetric Water Content (VWC); Accumulated solar radiation (AR).

**Figure 3 sensors-23-03177-f003:**
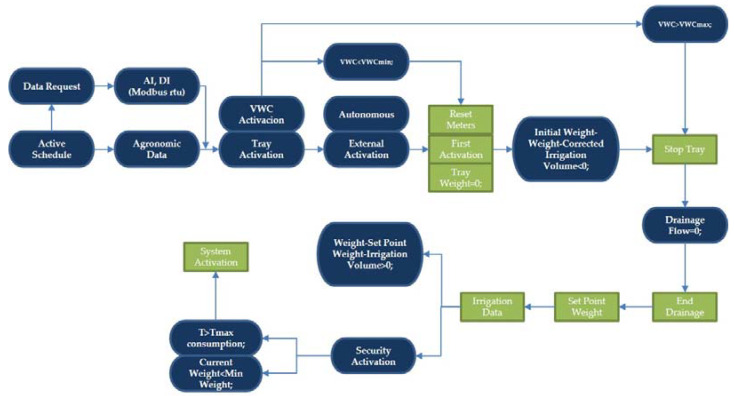
Grafcet of the process. In rounded blue: conditions, calculations, and data. In square green: actions.

**Figure 4 sensors-23-03177-f004:**
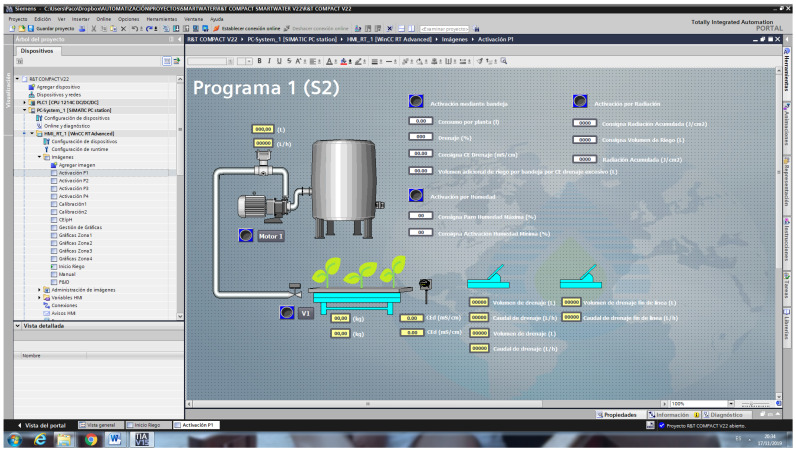
Screenshot of Human-Machine Interface of the SCADA system.

**Figure 5 sensors-23-03177-f005:**
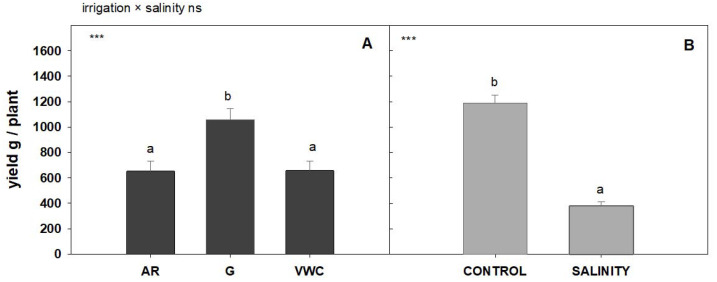
Average sum of commercial weights (gr/Plant) according to water quality and the irrigation method used. The error bar indicates the standard error of the mean ((**A**) n = 48 and (**B**) n = 72). Top left in this Figure means: “ns” indicates non-significant differences, and *** significant differences at *p* < 0.001. Values with different letters are significantly different at 95%, according to Duncan’s multiple range test.

**Figure 6 sensors-23-03177-f006:**
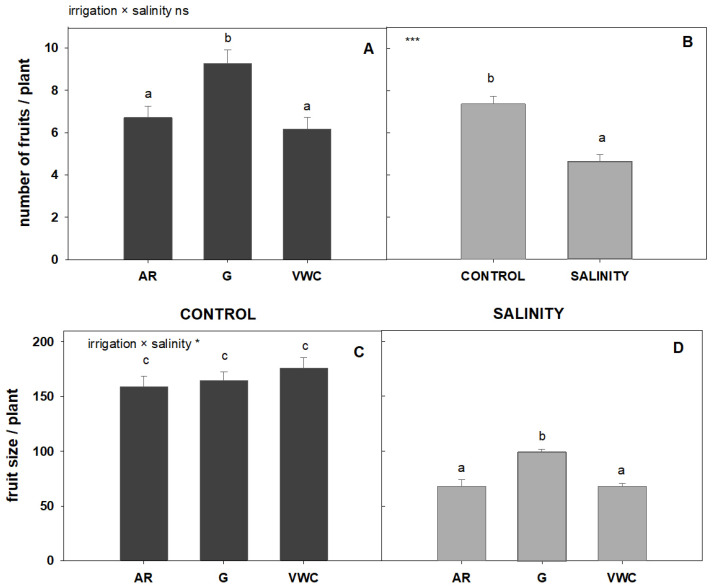
Average commercial number of fruits per plant (**A**,**B**) and average commercial fruit size (**C**,**D**), according to water quality and the irrigation used. The error bar indicates the standard error of the mean ((**A**) n = 48; (**B**) n = 72; (**C**) and (**D**) n = 24). Top left in this Figure means: “ns” indicates non-significant differences, * indicates significant differences at *p* < 0.05, and *** at *p* < 0.001. Values with different letters are significantly different at 95%, according to Duncan’s multiple range test.

**Figure 7 sensors-23-03177-f007:**
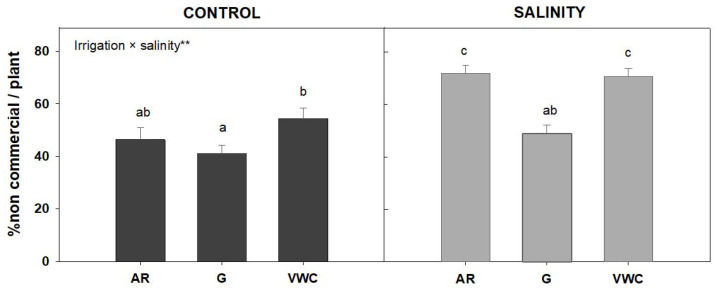
Average percentage of non-commercial yield (% non-commercial yield/plant), according to water quality and the irrigation used. The error bar indicates the standard error of the mean (n = 24). Top left in this Figure means: ** indicates significant differences at *p* < 0.05 between irrigation methods for each quality of water. Values with different letters are significantly different at 95%, according to Duncan’s multiple range test.

**Figure 8 sensors-23-03177-f008:**
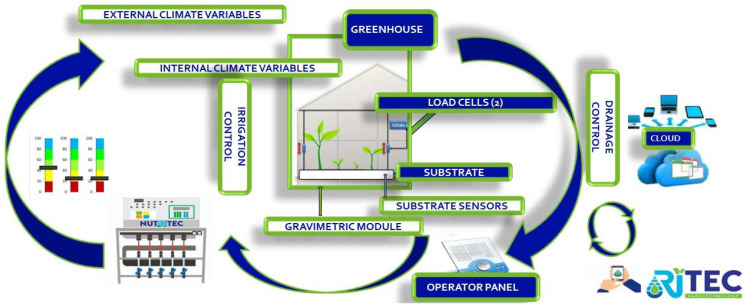
Gravimetric control system architecture. www.ritec.es (accessed on 10 March 2023).

**Figure 9 sensors-23-03177-f009:**
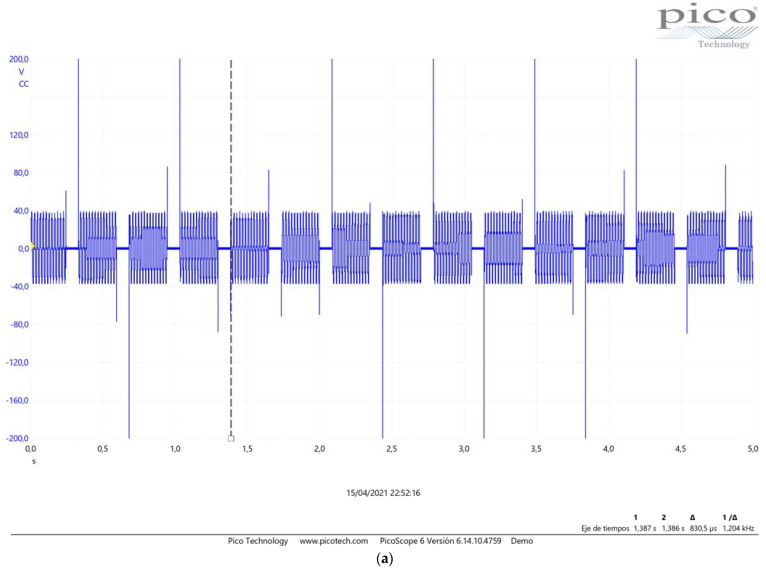
Voltage peaks (Vdc) in reed contact using intermediate relays (**a**), Voltage peaks (Vdc) in reed contact using solid state relays (**b**).

**Figure 10 sensors-23-03177-f010:**
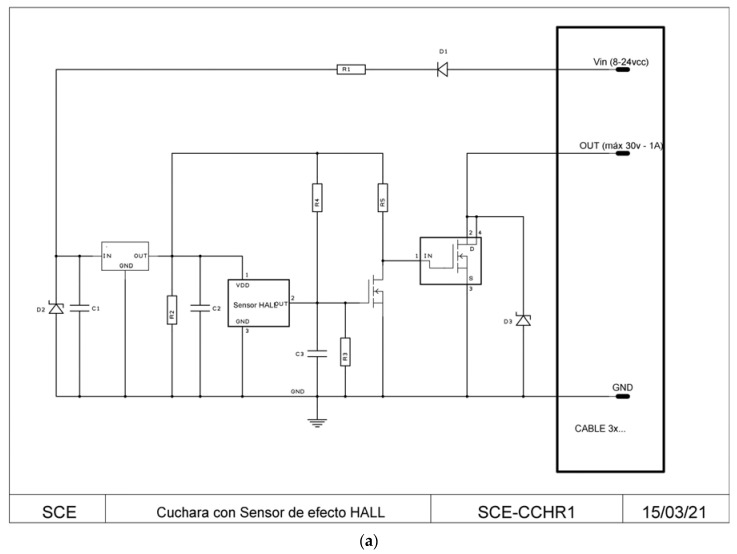
(**a**) Screenshot of diagram of double bucket meter with Hall effect sensor SCE-CCHR1. (**b**) Signal obtained by the most sensitive Hall effect sensor.

**Figure 11 sensors-23-03177-f011:**
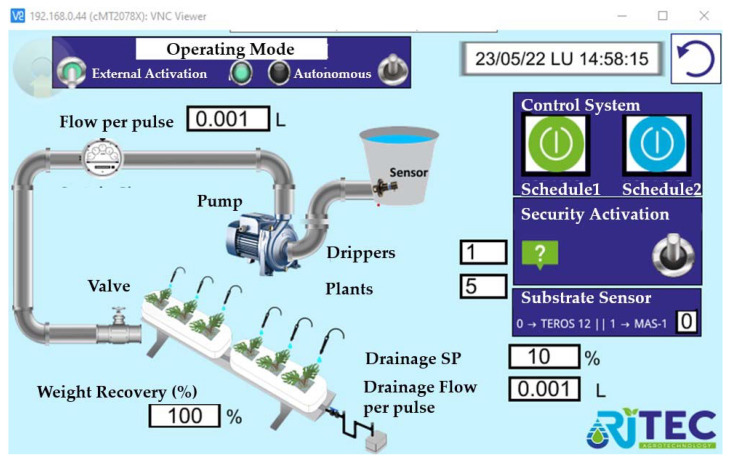
Screenshot example of HMI Operator’s panel.

**Figure 12 sensors-23-03177-f012:**
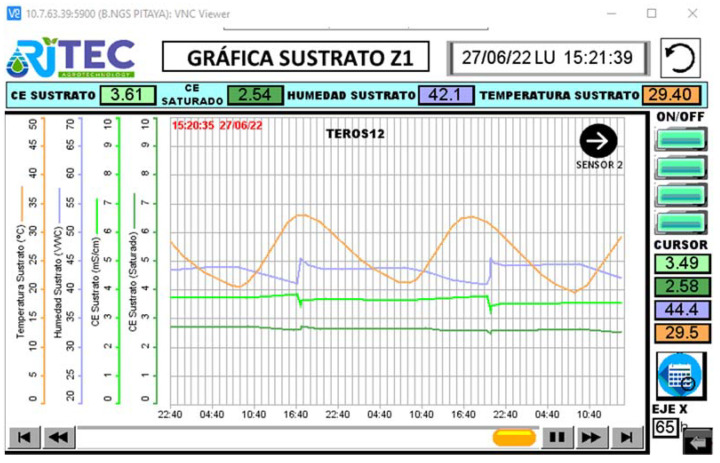
Screenshot of example of output data in the graphic format with substrate temperature, VWC, and EC.

**Figure 13 sensors-23-03177-f013:**
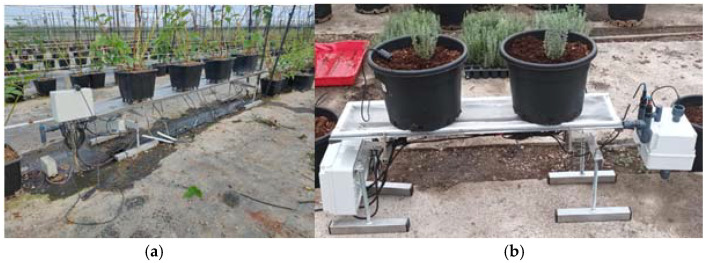
(**a**) Weighing tray with torsion load cells in a Mexican commercial fruit facility. (**b**) Small weighing tray in a commercial facility in the South of Spain.

**Table 1 sensors-23-03177-t001:** Weight average error and minimum detectable weight of the Weight System.

Error	Ascending Measures	Descending Measures
Ɛ%FS average	0.231	0.318
	**Near to Zero**	**Near to Full Scale**
**Min. detect. weight (gr)**	5	10

## Data Availability

Data sharing not applicable.

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
