# Peer review of "Development of Smart Irrigation Equipment for Soilless Crops Based on the Current Most Representative Water-Demand Sensors"

_sensors, 2023, doi:10.3390/s23063177_

Round 1
Reviewer 1 Report (New Reviewer)
General:
This paper is nicely drafted and easy to read.
Abstract:
Please clearly mention hypothesis, objective, and end abstract with future implications of this study.
Introduction:
Line 43, 56, 61: Please merge small paragraphs into bigger one.
Line 106: why it is in red fonts.
Line 135: Clearly highlight hypothesis, objectives, and end with future implications of this work after objectives.
M&M:
Line 147: Map with site of experiment will help readers.
PLEASE MENTION ABOUT FIELD EXPERIMENT DESIGN AND REPLICATES AND TERATMENTS.
Table 1 and 2 can be merged.
Results: merge small paragraphs
Figure 1, 2 and all: Improve figure resolution and use same font as text (Palatino linotype)
Line 238-24: Unbold, choose as MDPI text not as figure caption.
Few of the figures (11 and 12, 13) can go to supplementary.
Discussion:
I would suggest the authors to have a more supported discussion with references considering the main point: The limitations of method and considerations when to apply the studied methodology and then the potential next steps or further investigation to address these limitations.
Conclusion: Please merge small paragraphs and highlight about future implications and challenges of this study.
References: Please double check the style of references and missing one
Author Response
Please see attachment, including a new reviewed version of the paper.

Reviewer 2 Report (New Reviewer)
It is good to Develop Smart Irrigation Equipment for prevention of worsening the edaphoclimatic conditions. The study is needed. But theories and principles are not clearly introduced in this paper. I recommend authors to pay attention to this aspect.
•The abstract needs to be rewritten to point out significance and impact of the paper.
•Authors need to add some more current articles to improve the paper's overall quality. Comparative analysis of the current publications should be included.
•Paper needs to polish and provide a detailed explication of theoretical/systematic aspects behind.
•Notations and acronyms used in this paper should be summarized in a table.
•Improve the quality of figures and explain those properly.
•A proof-reading is highly suggested in order to correct wordings and typos.
Author Response
Please see attachment, including a new reviewed version of the paper.

Reviewer 3 Report (New Reviewer)
This manuscript has been improved and it is suggested that the manuscript be accepted.
Author Response
Thank you very much for your review and kind comments.
Please see attachement with the paper improved as a review from other reviewers.

This manuscript is a resubmission of an earlier submission. The following is a list of the peer review reports and author responses from that submission.
Round 1
Reviewer 1 Report
The article describes an automated system for irrigation control in soilless crops.
Introduction: The introduction sets out the characteristics of soilless crops and their advantages, but no analysis is made of the various systems analysed (144), determining that a new low-cost and scalable system is needed???? when??, where??
Materials and Methods: From a scientific point of view, it makes no sense to buy the 3 methods described, as the gravimetric method is considered a reference, i.e. none of the other two can adjust the irrigation needs as well as the gravimetric one, so such comparisons are of no scientific interest.
Discussion of results: Again, this is a poorly focused article from a scientific point of view. It does not contribute anything to compare 3 methods of irrigation control when you know that one of them is direct and the others are not and your discussion is to say that this is indeed the best.
Only section 3.1, which discusses the effect of the different irrigation methods on production, and the analysis of the results, is of some interest, although of low scientific rigour, since the conclusions of this section have been widely discussed in the literature and are therefore obvious and do not contribute anything new.
The detailed description of the components of the "winning" system, including the software and hardware, reinforces the conclusion that the work presented in this journal is not appropriate, it is an informative/descriptive article of a system for irrigation control in soilless cultivation, but it does not contribute anything from a scientific point of view.
Apart from the general appreciations, the work would need a great effort of formatting and correction, numerous acronyms without detail, equipment and devices without description, etc...
Author Response
Please see the attachment with the responses detailed and a new version of the paper with changes highlighted in red.

Reviewer 2 Report
This paper describes an experiment that compares three different control mechanisms for agriculture and demonstrates a comparison on a common tomato plant. In addition, it compares two types of water, one which is salinised.
The paper then goes on to describe the development of an alternative control system that implements the best strategy from part 1.
My main problem is that the paper is therefore neither one thing or the other. The initial experiment is interesting and draws some good conclusions about salinity. It draws other conclusions about which of the three techniques produces the best outcome. However, I find this part unconvincing as each system uses a separate proxy measurement with arbitrary thresholds for irrigation and so I think is an unfair comparison for production. Changing the thresholds may well produce different results for any of the systems tested, so concluding that the gravimetric method is best is therefore flawed. The authors may have other reasons for then going on to develop a bespoke system to support this method, but the conclusions drawn here are not strong enough to warrant it.
The later part of the paper is more of a product development discussion and although very solid engineering, will not be of interest to the audience interested in the performance of the different systems. We also have no conclusion at the end of the paper - it just finishes, so we don't know if the new control system works well in practice or not.
Therefore I feel that this paper is unsuitable for publication in its current form and should be broken into two papers - one on the experimental performance of the different measuring systems, which I found to be well done and interesting (the salinity results are particularly interesting) but would benefit from more detailed discussion and analysis of the results, and a second paper describing the development of the new control system, describing its novelty and performance benefits over the current solutions.
Author Response

(The authors gave the same response as above.)
